# LEARNING COVARIATE-SPECIFIC EMBEDDINGS WITH TENSOR DECOMPOSITIONS

## ABSTRACT

Word embedding is a useful approach to capture co-occurrence structures in a large corpus of text. In addition to the text data itself, we often have additional covariates associated with individual documents in the corpus—e.g. the demographic of the author, time and venue of publication, etc.—and we would like the embedding to naturally capture the information of the covariates. In this paper, we propose a new tensor decomposition model for word embeddings with covariates. Our model jointly learns a *base* embedding for all the words as well as a weighted diagonal transformation to model how each covariate modifies the base embedding. To obtain the specific embedding for a particular author or venue, for example, we can then simply multiply the base embedding by the transformation matrix associated with that time or venue. The main advantages of our approach is data efficiency and interpretability of the covariate transformation matrix. Our experiments demonstrate that our joint model learns substantially better embeddings conditioned on each covariate compared to the standard approach of learning a separate embedding for each covariate using only the relevant subset of data. Furthermore, our model encourages the embeddings to be "topic-aligned" in the sense that the dimensions have specific independent meanings. This allows our covariate-specific embeddings to be compared by topic, enabling downstream differential analysis. We empirically evaluate the benefits of our algorithm on several datasets, and demonstrate how it can be used to address many natural questions about the effects of covariates.

## 1 INTRODUCTION

The use of factorizations of co-occurrence statistics in learning low-dimensional representations of words is an area that has received a large amount of attention in recent years, perhaps best represented by how widespread algorithms such as GloVe (Pennington et al., 2014) and Word2Vec (Mikolov et al., 2013) are in downstream applications. In particular, suppose we have a set of words $i \in [n]$, where $n$ is the size of the vocabulary. The aim is to, for a fixed dimensionality $d$, assign a vector $v_i \in \mathbb{R}^d$ to each word in the vocabulary in a way that preserves semantic structure.

In many settings, we have a corpus with additional covariates on individual documents. For example, we might have news articles from both conservative and liberal-leaning publications, and using the same word embedding for all the text can lose interesting information. Furthermore, we suggest that there are meaningful semantic relationships that can be captured by exploiting the differences in these conditional statistics. To this end, we propose the following two key questions that capture the problems that our work addresses, and for each, we give a concrete motivating example of a problem in the semantic inference literature that it encompasses.

**Question 1:** How can we leverage conditional co-occurrence statistics to capture the effect of a covariate on word usage?

For example, did William Shakespeare truly write all the works credited to him, or have there been other "ghostwriters" who have contributed to the Shakespeare canon? This is the famous Shakespeare authorship question, for which historians have proposed various candidates as the true authors of particular plays or poems (Hope, 1994). If the latter scenario is the case, what in particular distinguishes the writing style of one candidate from another, and how can we infer who the most likely author of a work is from a set of candidates?

**Question 2:** Traditional factorization-based embedding methods are rotationally invariant, so that individual dimensions do not have semantic meaning. How can we break this invariance to yield a model which aligns topics with interpretable dimensions?

There has been much interest in the differences in language and rhetoric that appeal to different demographics. For example, studies have been done regarding "ideological signatures" specific to voters by partisan alignment (Robinson et al.) in which linguistic differences were proposed along focal axes, such as the "mind versus the body" in texts with more liberal or conservative ideologies. How can we systematically infer topical differences such as these between different communities?

Questions such as these, or more broadly covariate-specific trends in word usage, motivated this study. Concretely, our goal is to provide a general framework through which embeddings of sets of objects with co-occurrence structure, as well as the effects of conditioning on particular covariates, can be learned jointly. As a byproduct, our model also gives natural meaning to the different dimensions of the embeddings, by breaking the rotational symmetry of previous embedding-learning algorithms, such that the resulting vector representations of words and covariates are "topic-aligned".

**Previous Work**   Typically, algorithms for learning embeddings rely on the intuition that some function of the co-occurrence statistics is *low rank*. Studies such as GloVe and Word2Vec proposed based on minimizing low-rank approximation-error of nonlinear transforms of the co-occurrence statistics. let $A$ be the $n \times n$ matrix with $A_{ij}$ the co-occurrence between words $i$ and $j$, where co-occurrence is defined as the (possibly weighted) number of times the words occur together in a window of fixed length. For example, GloVe aimed to find vectors $v_i \in \mathbb{R}^d$ and biases $b_i \in \mathbb{R}$ such that the loss

$$J(v, b) = \sum_{i,j=1}^{n} f(A_{ij})(v_i^T v_j + b_i + b_j - \log A_{ij})^2 \tag{1}$$

was minimized, where $f$ was some fixed increasing weight function. Word2Vec aimed to learn vector representations via minimizing a neural-network based loss function.

A related embedding approach is to directly perform principal component analysis on the PMI (pointwise mutual information) matrix of the words (Bullinaria & Levy). PMI-factorization based methods aim to find vectors $\{v_i\}$ such that

$$v_i^T v_j \approx PMI(A)_{ij} = \log \frac{\mathbb{P}(i, j)}{\mathbb{P}(i)\mathbb{P}(j)} \tag{2}$$

where the probabilities are taken over the co-occurrence matrix. This is essentially the same as finding a low-rank matrix $V$ such that $V^T V \approx PMI$, and empirical results show that the resulting embedding captures useful semantic structure.

The ideas of several previous studies on the geometry of word embeddings was helpful in formulating our model. A random-walk based mathematical framework for understanding these different successful learning algorithms was proposed (Arora et al., 2016), in which the corpus generation process is a random process driven by the random walk of a discrete-time discourse vector $c_t \in \mathbb{R}^d$. In this framework, our work can be thought of as analyzing the effects of covariates on the random walk transition kernel and the stationary distribution. Additionally, there have been previous studies of "multi-sense" word embeddings (Reisinger & Mooney, 2010) (Neelakantan et al., 2015), which is similar to our idea that the same word can have different meanings in different contexts. However, in the multi-sense setting, the idea is that the word intrinsically has different meanings (for example, "crane" can be an action, a bird, or a vehicle), whereas in ours, the different meanings are imposed by conditioning on a covariate. Finally, tensor methods have been used in other settings recently, such as collaborative filtering (Xiong et al., 2010) and (Li & Farias), to learn the effects of conditioning on some summary statistics.

**Our Contributions**   There are several reasons why a joint learning model based on tensor factorization is more desirable than performing GloVe $m$ times, where $m$ is the number of covariates, so

that each covariate-specific corpus has its own embedding. Our main contributions are a decomposition algorithm that addresses these issues, and the methods for systematic analysis we propose.

The first issue that arises is sample complexity. In particular, because for the most part words are used in roughly similar ways across different contexts, the resulting embeddings should not be *too different*, except perhaps along specific dimensions. Thus, it is better to jointly train an embedding model along the covariates to aggregate the co-occurrence structure, especially in cases where the entire corpus is large, but many conditional corpora (conditioned on a covariate) are small.

Secondly, simply training a different embedding for each corpus makes it difficult to compare the embeddings across the covariate dimension. Because of issues such as rotation invariance of GloVe-like models, specific dimensions mean different things across different runs (and initializations) of these algorithms. The model we propose has the additional property that it induces a natural basis to view the embeddings in, one which is "topic-aligned" in the sense that it is not rotation-invariant and thus implies independent topic meanings given to different dimensions.

**Paper Organization**   In section 2, we provide our embedding algorithm, as well as mathematical justification for its design. In section 3, we detail our dataset. In section 4, we validate our algorithm with respect to intrinsic properties and standard metrics. In section 5, we propose several experiments for systematic downstream analysis.

## 2   TENSOR COVARIATE EMBEDDING ALGORITHM

**Notation**   Throughout this section, we will assume a vocabulary of size $n$ and a discrete covariate to condition on of size $m$ (for example, the community that the corpus comes from, i.e. liberal or conservative discussion forums). It is easy to see how our algorithm generalizes to higher-order tensor decompositions when there are multiple dimensions covariates to condition on (for example, slicing along community and slicing along timeframe simultaneously). Words will be denoted with indices $i, j \in [n]$ and covariates with index $k \in [m]$. Dimensions in our embedding are referred to by index $t \in [d]$.

We will denote the co-occurrence tensor as $A \in \mathbb{R}^{n \times n \times m}$, where $A_{ijk}$ denotes how many times words $i$ and $j$ occurred together within a window of some fixed length, in the corpus coming from covariate $k$. The result of our algorithm will be two sets of vectors, $\{v_i \in \mathbb{R}^d\}$ and $\{c_k \in \mathbb{R}^d\}$, as well as bias terms that also fit into the objective. Finally, let $\odot$ denote the element-wise product between two vectors.

**Objective Function and Discussion**   Here, we give the objective function our method minimizes, and provide some explanation for how one should imagine the effect of the covariate weights. The objective function we minimize is the following *partial non-negative tensor factorization* objective function for jointly training word vectors and weight vectors representing the effect of covariates, adapted from the original GloVe objective (note that $c_k \odot v_i = \mathbf{diag}(c_k)v_i$, where $\mathbf{diag}(c_k)$ is the diagonal matrix weighting of covariate $k$):

$$J(v, c, b) = \sum_{i,j=1}^{n} \sum_{k=1}^{m} f(A_{ijk})((c_k \odot v_i)^T (c_k \odot v_j) + b_{ik} + b_{jk} - \log A_{ijk})^2 \qquad (3)$$

which is to be optimized over $\{v_i \in \mathbb{R}^d\}$, $\{c_k \in \mathbb{R}^d\}$, and $\{b_{ik} \in \mathbb{R}\}$. To gain a little more intuition for why this is a reasonable objective function, note that the resulting objective for a single "covariate slice" is essentially

$$J_k(v, c, b) = \sum_{i,j=1}^{n} f(A_{ijk})((c_k \odot v_i)^T (c_k \odot v_j) + b_{ik} + b_{jk} - \log A_{ijk})^2 \qquad (4)$$

which fits the vectors $c_k \odot v_i$ to the data, thus approximating the statistic $\log A_{ijk}$ with $\sum_{t=1}^{d} v_{it} v_{jt} c_{kt}^2$. Note that in the case $m = 1$, the model we use is identical to the standard GloVe

model since the $c_k$ can be absorbed into the $v_i$. We used $f(x) = (\frac{\min(100,x)}{100})^{0.75}$, to parallel the original objective function in (Pennington et al., 2014).

One can think of the dimensions our model learns as independent topics, and the effects of the covariate weights $c_k$ as upweighting or downweighting the importance of these topics in contributing to the conditional co-occurrence statistics.

**A Geometric View of Embeddings and Tensor Decomposition**   Here we provide a geometric perspective on our model in the context of some prior work. This geometric interpretation is not necessary to applying our method and to understand its results. Throughout this section, note that at a high level, the aim of our method is to learn sets $\{v_i \in \mathbb{R}^d\}$, $\{c_k \in \mathbb{R}^d\}$, and $\{b_{ik} \in \mathbb{R}\}$ for $1 \leq i \leq n, 1 \leq k \leq m$, such that for a fixed $k$, the vectors $\{c_k \odot v_i\}$ and the biases $\{b_{ik}\}$ approximate the vectors and biases that would have been learned from running GloVe on only the $k^{th}$ slice of the co-occurrence tensor.

We now provide some rationale for why this is a reasonable objective. The main motivation for our algorithm is the following geometric intuition inspired in part by the model of Arora et al. (2016). In their model, corpus generation is determined by the nature of the random walk performed by a context vector $c_t$ over time steps. In particular, for each timestep $t$, the context vector $c_t$ emits words $i$ with probability $\propto \exp(v_i \cdot c_t)$, and then $c_t$ updates its location based on a transition distribution. The assumption is that the transition distribution has a stationary distribution that is uniform over some ellipse. One natural equivalence family of transition matrices that preserves this property is the group resulting from multiplication by a positive semidefinite matrix.

Therefore, we consider a natural extension of this model, where the embedding resulting from conditioning on different covariates is equivalent to multiplying the transition matrix by a symmetric PSD matrix. Alternatively, this is equivalent to a model where the transition kernel remains unchanged, but the embedding vectors themselves are multiplied by a symmetric PSD matrix (namely, the Moore-Penrose pseudoinverse of the original PSD matrix). This is the viewpoint we adopt.

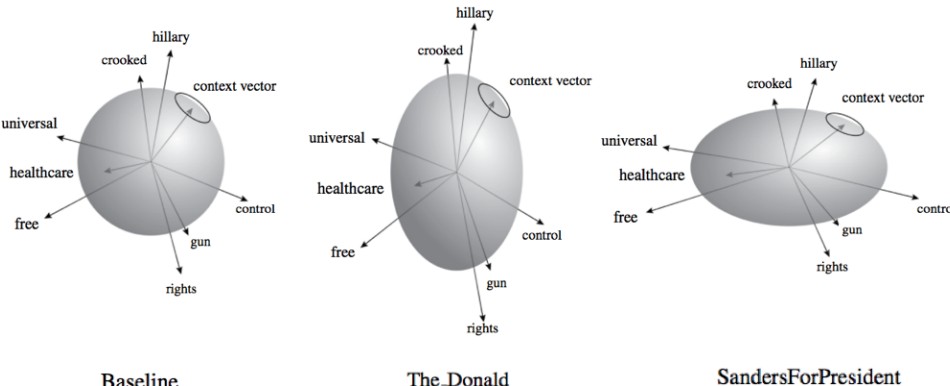

Figure 1: The effects of conditioning on covariates (covariates are discussion forums, described in Section 3). Left: baseline embedding with some possible word embedding positionings. Middle, right: embedding under effect of covariates: for example, "hillary" and "crooked" are pushed closer together under effects of *The_Donald*, and "healthcare" pushed closer to "universal" and "free" under effects of *SandersForPresident*. Context vectors and random walk transitions also shown.

In this framework, assign each covariate $k$ its own symmetric PSD matrix, $B_k$. It is well-known that any symmetric PSD matrix can be factorized as $B_k = R_k^T D_k R_k$ for some orthonormal basis $R_k$ and some (nonnegative) diagonal $D_k$. Thus it suffices to consider the effect of a covariate on some ground truth "base embedding" $M$ as applying the linear operator $B_k$ to each embedding vector, resulting in the new embedding $B_k M$.

This model is quite expressive in its own right, but we consider a natural restriction where we propose that there exists some universal (at least across covariates) basis $R$ under which the resulting

embeddings are affected by covariates via multiplication by just a diagonal matrix, instead of a PSD matrix. In particular, we note that

$$(B_k M)^T (B_k M) = M^T R_k^T D_k R_k R_k^T D_k R_k M = M_k'^T D_k^2 M_k' \tag{5}$$

where $M_k' = R_k M$ is a rotated version of $M$. Now, in the restricted model where all the $R_k$ are equal, we can write all the $M_k'$ as $RM$, so it suffices to just consider the rotation of the basis that the original embedding was trained in where $R$ is just the identity (since matrix-factorization based word embedding models are rotation invariant). Under this model, the co-occurrence statistics under some transformation should be equivalent to $M^T D_k^2 M$. A careful comparison shows that this approximation is precisely that which is implied by equation 4, as desired.

Note that this is essentially saying that in this distributed word representation model, there exists some rotation of the embedding space under which the effect of the covariate separates along dimensions. The implication is that there are some set of independent "topics" that each covariate will upweight or downweight in importance (or possibly ignore altogether with a weight of 0), characterizing the effect of this conditioning directly in terms of the effect on these topics.

**Algorithm Details**   Our model learns the resulting parameters $\{v_i \in \mathbb{R}^d\}$, $\{c_k \in \mathbb{R}^d\}$, and $\{b_{ik}\}$, by using the Adam (Kingma & Ba, 2014) algorithm, which was empirically shown to yield good convergence results in the original GloVe setting. The specific hyperparameters used for each dataset will be described in the next section. The word and covariate weight vectors were initialized as random unit vectors [1].

## 3 DATASET

We evaluated our method in to primary datasets. In both datasets, co-occurrence statistics were formed by considering size 8 windows and using an inverse-distance weighting (e.g. neighboring words had 1 added to their co-occurrence, and words 3 apart had $\frac{1}{3}$ added), which was suggested by some implementations of (Pennington et al., 2014).

The first dataset, referred to as the "book dataset", consists of the full text from 29 books written by 4 different authors. The books we used were J.K. Rowling's "Harry Potter" series (7 books), "Cormoran Strike" series (3 books), and "The Casual Vacancy"; C. S. Lewis's "The Chronicles of Narnia" series (7 books), and "The Screwtape Letters"; George R. R. Martin's "A Song of Ice and Fire" series (5 books); and Stephenie Meyer's "Twilight" series (4 books), and "The Host". These books are fiction works in similar genres, with highly overlapping vocabularies and common themes. A trivial way of learning series-specific tendencies in word usage would be to cluster according to unique vocabularies (for example, only the "Harry Potter" series would have words such as "Harry" and "Ron" frequently), so the co-occurrence tensor was formed by looking at all words that occurred in all of the series with multiple books, which eliminated all series-specific words. Furthermore, series by the same author had very different themes, so there is no reason intrinsic to the vocabulary to believe the weight vectors would cluster by author. The vocabulary size was 5,020, and after tuning our algorithm to embed this dataset, we used 100 dimensions and a learning rate of $10^{-5}$.

The second dataset, referred to as the "politics dataset", was a collection of comments made in 2016 in 6 different subreddits on the popular discussion forum reddit, and was selected to address both Questions 1 and 2. The covariate was the discussion forum, and the subreddits we used were *AskReddit*, *news*, *politics*, *SandersForPresident*, *The_Donald*, and *WorldNews*. *AskReddit* was chosen as a baseline discussion forum with a very general vocabulary usage, and the discussion forums for the Sanders and Trump support bases were also selected, as well as three politically-relevant but more neutral communities (it should be noted that the *politics* discussion forum tends to be very left-leaning). We considered a vocabulary of size 15,000, after removing the 28 most common words (suggested by other works in the embedding literature) and entries of the cooccurrence tensor with

---

[1]We also experimented with initializing covariate weight vectors as random vectors centered around the all 1 vector. This initialization also yielded the sparsity patterns discussed in the next section, but converged at a slower rate, and performed similarly on downstream metrics as initializing near all 0, so we kept this initialization.

less than 10 occurrences (for the sake of training efficiency). The embedding used 200 dimensions and a learning rate of $10^{-5}$.

# 4 EXPERIMENTAL VALIDATION

## 4.1 CLUSTERING BY WEIGHTS

We performed the tensor decomposition algorithm on the book dataset, and considered how well the weight vectors of the covariate clustered by series and also by author.

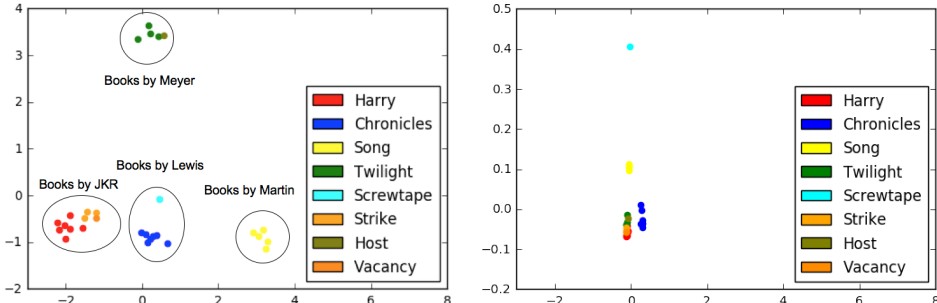

(a) 2D PCA of book dataset weight vectors (tensor decomposition algorithm, 100 dimensions)

(b) 2D PCA of book dataset topic vectors, as predicted by LDA (100 topics)

For every book in every series, the closest weight vectors by series were all the books in the same series. Furthermore, for every book written by every author, the closest weight vectors by author were all the books by the same author. This clear clustering behavior even when only conditioning on co-occurrence statistics for words that appear in all series (throwing out series-specific terms) implies the ability of the weight vectors to cluster according to higher-order information, perhaps such as writing style. As a baseline, we considered the topic breakdown on the same co-occurrence statistics predicted by Latent Dirichlet Allocation (Blei et al., 2003) with various dimensionalities (20, 50, 100), all of which failed to produce any meaningful clusters.

## 4.2 DATA EFFICIENCY AND VALIDATION

Consider the problem of learning an embedding for the text of a particular book (or series). The main advantage given by using the covariate-specific embedding that our method yields over applying GloVe to the individual slice is a data efficiency - by pooling the co-occurrence statistics of words across other books, we are able to give better (less noisy) estimates of the vectors, especially for sparsely-occurring or nonexistent words. For reference, individual books contained between 26747 and 355814 words.

To this end, we performed the following experiment. For some book $k$, consider two embeddings: 1) the result of performing GloVe on the co-occurrence statistics of just the book, and 2) the (weighted) embedding resulting from our algorithm, specific to the book. Then, we tested these resulting embeddings using a standard suite of evaluation metrics, including cluster purity and correlation similarities. Our method outperformed method 1 on all 7 tasks, often by a significant margin.

Table 1: Standard metric performance, GloVe vs. our model, averaged over 7 Harry Potter books

|  | Cluster purity | | | Correlation similarities | | | |
|---|---|---|---|---|---|---|---|
|  | AP | BLESS | Battig | MEN | MTurk | RG65 | RW |
| tensor | 0.1602 | 0.2185 | 0.0877 | 0.1370 | 0.0719 | 0.1272 | 0.0902 |
| GloVe | 0.1297 | 0.2042 | 0.0770 | 0.0593 | 0.0341 | 0.0133 | 0.0588 |

### 4.3 Sparsity of Weight Vectors

Because of experimentally verified isotropic distributional properties of word embedding vectors (Arora et al., 2016), it is unreasonable to ask for sparsity in the word embeddings themselves. However, our topic-specific weighting scheme for the covariate weight vectors implies that sparsity is desirable in the weights. The weight sparsity resulting from our algorithm was experimentally verified through many runs of our embedding method, as well as across different optimization methods.



Figure 3: Histogram, sizes of weights in covariate vectors for politics dataset. Smallest bucket is zero ($< 10^{-10}$). Sparsity of covariates in book dataset deferred to appendix.

Note that in the objective function (3), sparse coordinates will become "stuck" at 0, because the gradient update of $c_{kt}$ is proportional to $c_{kt}$:

$$\frac{\partial J}{\partial c_{kt}} = \sum_{i,j=1}^{n} 4f(A_{ijk})((c_k \odot v_i)^T(c_k \odot v_j) + b_{ik} + b_{jk} - \log A_{ijk})c_{kt} \qquad (6)$$

The dimensions that the covariates were sparse in did not overlap by much: the average number of sparse coordinates per weight vector was 20.7, and the average number of coordinates that was sparse in both vectors of a random pair was 5.2. This suggest that the set of "topics" each conditional slice of the tensor does not care about is fairly specific. Experimenting with regularizing the word vectors forced the model into a smaller subspace (i.e. sparse dimensions existed but were shared across all the words), which is not useful and provides further evidence for the natural isotropic distribution of word vectors. Regularizing the weights of the covariate vectors by adding an $l_1$ penalty term did not noticeably change the sparsity pattern.

Table 2: Number of sparse coordinates in covariate weight vectors, 5 runs.

| Statistics | AskReddit | news | politics | SandersForPresident | The_Donald | worldnews |
|---|---|---|---|---|---|---|
| mean | 18.2 | 16.8 | 25.6 | 36.8 | 27.2 | 23.2 |
| std | 1.4 | 2.4 | 2.2 | 3.6 | 2.6 | 3.1 |

To confirm that the sparsity was a result of separation of covariate effects rather than an artifact of our algorithm, we ran our decomposition on a co-occurrence tensor which was the result of taking the same slice (subreddit) and subsampling its entries 3 times, creating 3 slices of essentially similar co-occurrence statistics. We applied our algorithm with different learning parameters, and the resulting weight vectors after 90 iterations (when the outputs vectors converged) were extremely non-sparse, with between 0 and 2 sparse coordinates per weight vector. The dimensions that are specifically 0 for a covariate corresponds to topics that are relatively less relevant for that covariate. In the next section, we develop methods to systematically interpret the covariate weights in terms of topics.

## 5 Interpretation

### 5.1 Inference of Topic Meaning

A simple test of inferring topic meaning (i.e. topics coordinates are associated with) is to consider the set of words which are large in the given coordinate. Concretely the task is, given some index

$t \in [d]$, output the words whose (normalized) vectors have largest value in dimension $t$. We show the results of this experiment for several of the sparse coordinates in the *AskReddit* weight vector:

Table 3: Top words for selected dimensions: topic inference task

| $t$ | Most heavily weighted words, dimension $t$ | Meaning |
|---|---|---|
| 99 | horses, texts, cat, teenager, ancient, animals, grandma, coworkers | people, animals |
| 120 | tables, driveway, designing, customer, memories, stations | domestic, jobs |
| 183 | gate, territory, backpack, directions, outlets, nations, phillipines | foreign relations |
| 194 | shallow, sweat, coastal, disciplined, beliefs, tricks, bastion, marines | military |

There are several conclusions to be drawn from this experiment. Firstly, while there is some clear noise, specific topic meanings do seem to appear in certain coordinates (which we infer in the table above). It is reasonable that meaning would appear out of coordinates which are sparsely weighted in some covariate, because this means that it is a topic that is relevant in other discussion forums but purposely ignored in some specific forum, so it is likely to have a consistent theme. When we performed this task for coordinates which had low variance in their weights between covariates, the resulting words were much less consistent.

It is also interesting to see how covariates weight a topic whose meaning we have identified. For example, for coordinate 194 (corresponding to military themes), *AskReddit* placed negligible weight, *news* and *worldnews* placed weight 2.06 and 2.04 respectively, *SandersForPresident* and *The_Donald* placed weight 0.41 and 0.38 respectively, and *politics* placed weight 0.05. This process can also be reversed - for example, consider coordinates small in *worldnews* and large in *news*. One example was coordinate 188, and upon checking the words that were large in this coordinate ({taser, troops, supremacists, underpaid, rioters, amendment, racially, hispanic}) it seemed clear that it had themes of rights and protests, which makes sense as a domestic issue, not a global one.

## 5.2 TOPICAL WORD DRIFT

We performed the following experiment: which pairs of words start off close in the baseline embedding, yet under some covariate weights move far apart (or vice versa)? Concretely, the task is, for a fixed word $i$ and a fixed covariate $k$, identify words $j$ such that $||c_k \odot v_i - c_k \odot v_j|| \gg ||v_i - v_j||$ or $||c_k \odot v_i - c_k \odot v_j|| \ll ||v_i - v_j||$, where the magnitude of drift is quantified by the ratio of distances in the normalized embedding. The motivation is to find words whose general usage is similar, but have very different meanings in specific communities. We present a few representative examples of this experiment below, for $k$ = *The_Donald*.

Table 4: Representative examples of word movement, drift task

| Word $i$ | Drift direction | Words with strongest drift |
|---|---|---|
| hillary | Move closer | crooked, lying, shillary, killary, (various other slurs) |
| hillary | Move further | electable, compromise, favored, united, contributions |
| gun | Move closer | merchandise, milo, flair, blockbuster, sentiments, fakenews |
| gun | Move further | assault, child, fanatics, affecting, drug, policy, problem, police |
| immigrant | Move closer | interrogation, unauthorized, uninsured, parasite, (various other slurs) |
| immigrant | Move further | child, creates, affecting, abused, future, policy, attorneys, protect |

Combining the previous two sections allows us to do an end-to-end case study on words that drift under a covariate, so we can explain specifically which topics (under reweighting) caused this shift. For example, the words "immigrant" and "parasite" were significantly closer under the weights placed by *The_Donald*, so we considered dimensions that were simultaneously large in the vector $v_{immigrant} - v_{parasite}$ and sparse in the weight vector $c_{The\_Donald}$. The dimensions 89 and 139 were sparse and also the 2nd and 3rd largest coordinates in the difference vector, so they had a large contribution to the subspace which was zeroed out under the reweighting. Words that were large in these dimensions (and thus representative of the zeroed out subspace meaning) included {misunderstanding, populace, scapegoat, rebuilding} for 89, and {answers, barriers, applicants,

backstory, selfless, indigenous} for 139. This suggests two independent reasons for the drift, namely dimensions corresponding to emotional appeal and legal immigration respectively being zeroed out.

## 5.3 COVARIATE-SPECIFIC ANALOGIES

One of the most famous downstream applications of recent embedding methods such as (Pennington et al., 2014) and (Mikolov et al., 2013) is their ability to capture analogies. This is typically formulated as $a : b :: c : d \leftrightarrow v_a - v_b \approx v_c - v_d$, for example $v_{woman} - v_{queen} \approx v_{man} - v_{king}$. We considered how well our method captured *covariate-specific analogies*, which appear in a covariate-specific embedding but not most others. To this end, we considered experiments of the form: for fixed words $a, b, c$, determine words $d$ such that for some covariate $k$, the quantity

$$\frac{(c_k \odot v_a - c_k \odot v_b) \cdot (c_k \odot v_c - c_k \odot v_d)}{||c_k \odot v_a - c_k \odot v_b|| ||c_k \odot v_c - c_k \odot v_d||} \tag{7}$$

is small, yet for other $k$, the quantity is large. The intuition is that under the covariate transform, $v_c - v_d$ points roughly in the same direction as $v_a - v_b$, and $d$ is close to $c$ in semantic meaning.

In particular, we set $a$ = "hillary", $c$ = "trump", and found words $b$ for which there existed a $d$ consistently at the top across subreddits (implying existence of strong analogies). For example, when $b$ = "woman", $d$ = "man" was the best analogy for every weighting. Then, for these $b$, we considered words $d$ whose relative rankings in the subreddits had high variance. The differential analogies captured were quite striking, and we present several representative examples in Table 5.

Table 5: Analogies task. Each best analogy $d$ was one of the top-ranked words in every embedding. We present words whose "relative analogy" rank was enriched in some embedding. Subreddits are color-coded: green for news-related WN and N (*worldnews*, *news*), blue for left-leaning P and S (*politics*, *SandersForPresident*), red for right-leaning D (*The_Donald*), black for A (*AskReddit*).

| Word $b$ | Best analogy $d$ | Word | High rank | Low rank |
|---|---|---|---|---|
| woman | man | abysmal | 1351 (S), 2218 (P) | 14329 (base), 14077 (D) |
| | | amateur | 1543 (P), 3966 (S) | 13840 (base), 13734 (D) |
| | | zionist | 1968 (P), 2327 (S) | 14173 (base), 14248 (A) |
| | | politician | 2796 (WN), 3155 (D) | 11959 (base), 10386 (S) |
| | | president | 2452 (D), 3564 (WN) | 12257 (base) |
| | | nationalists | 208 (S), 606 (D) | 8916 (base), 7526 (A) |
| democrat | republican | south | 3511 (P) | 11091 (base) |
| | | bigot | 400 (S), 530 (A) | 12888 (D), 12994 (WN) |
| liberal | conservative | christian | 33 (P) | 12756 (D), 12722 (WN) |
| | | white | 619 (P) | 12824 (D), 13273 (base) |
| | | racist | 252 (P) | 12930 (S), 12779 (D) |
| | | sociopathic | 1756 (D) | 13744 (P), 13389 (A) |
| | | disenfranchised | 3693 (D) | 11267 (base) |
| politician | businessman | confidence | 2768 (D) | 10528 (base) |
| | | questionable | 598 (WN) | 13002 (base) |
| | | irrational | 2153 (N), 3430 (P) | 13305 (base) |

**Discussion** We have presented a joint tensor model that essentially learns an embedding for each word and for each covariate. This model makes it very simple to compute the covariate specific embedding: we just take the element-wise vector product. It also enables us to systematically interpret the covariate vector by looking at dimensions along which weight is large or 0. Our experiments show that these dimensions can be interpreted as coherent topics. While we focus on word embeddings, our tensor covariate embedding model can be naturally applied in other settings. For example, there is a large amount of interest in learning embeddings of individual genes to capture biological interactions. The natural covariates here are the cell types and our method would be able to model cell-type specific gene interactions. Another interesting setting with conditional covariates would be time-series specific embeddings, where data efficiency becomes more of an issue. We hope our framework is general enough that it will be of use to practitioners in these settings and others.

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

## A    SPARSITY RESULTS: BOOK DATASET

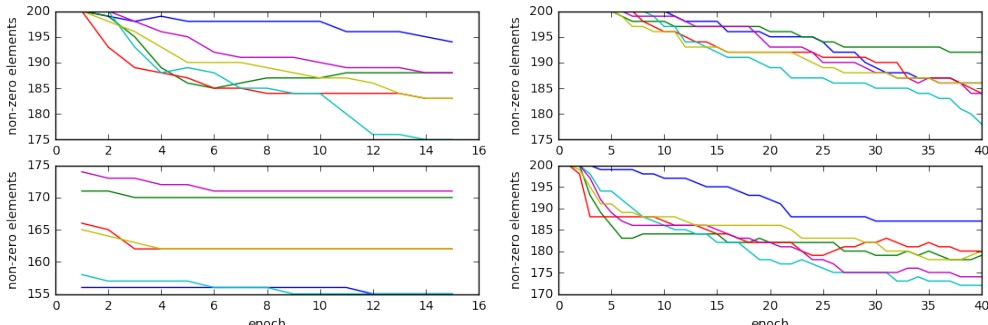

Figure 4: Histogram, sizes of weights in covariate vectors of book data.

Number of sparse coordinates (out of 100) were as follows, by series and then book order: Harry (0, 5, 1, 3, 7, 4, 0), Chronicles (0, 0, 0, 1, 0, 0, 0), Song (8, 8, 9, 4, 11), Twilight (6, 6, 8, 7), Screwtape (5), Strike (3, 2, 2), Host (6), Vacancy (4).

## B    ALGORITHM SETTING NOTES

(a) Non-zero dimensions in weight vectors by epoch and optimization method. Upper: Adam; lower: Adagrad.

(b) Non-zero dimensions in weight vectors by epoch. Upper: initialization centered around all-1 vector; lower: centered around all-0 vector.

We also experimented with using (Duchi et al.) as the optimization method, but the resulting weight vectors in the politics dataset had highly-overlapping sparse dimensions. This implies that the optimization method tried to fit the model to a smaller-dimensional subspace, which is not a desirable source of sparsity.

