# OpenReview forum: "Learning Covariate-Specific Embeddings with Tensor Decompositions"
_ICLR.cc/2018/Conference — Reject_

### Official Review · AnonReviewer3 · 2017-11-21
**Good idea, not well evaluated against other methods**

**Rating:** 5
**Confidence:** 3

**Review:**

This paper produces word embedding tensors where the third order gives covariate information, via venue or author. The model is simple: tensor factorization, where the covariate can be viewed as warping the cosine distance to favor that covariate's more commonly cooccuring vocabulary (e.g. trump on hillary and crooked)


There is a nice variety of authors and words, though I question if even with all those books, the corpus is big enough to produce meaningful vectors. From my own experience, even if I spend several hours copy-pasting from project gutenberg, it is not enough for even good matrix factorization embeddings, much less tensor embeddings. It is hard to believe that meaningful results are achieved using such a small dataset with random initialization.

I think table 5 is also a bit strange. If the rank is > 1000 I wonder how meaningful it actually is. For the usual analogies task, you can usually find what you are looking for in the top 5 or less.

It seems that table 1 is the only evaluation of the proposed method against any other type of method (glove, which is not a tensor-based method). I think this is not sufficient.

Overall, I believe the idea is nice, and the initial analysis is good, but I think the evaluation, especially against other methods, needs to be stronger. Methods like neelakantan et al's multisense embedding, for example, which the work cites, can be used in some of these evaluations, specifically on those where covariate information clearly contributes (like contextual tasks). The addition of one or two tables with either a standard task against reported results or created tasks against downloadable contextual / tensor embeddings would be enough for me to change my vote.

---

### Official Review · AnonReviewer1 · 2017-11-26
**nice work, see suggested improvements**

**Rating:** 5
**Confidence:** 5

**Review:**

The authors present a method for learning word embeddings from related groups of data. The model is based on tensor factorization which extends GloVe to higher order co-ocurrence tensors, where the co-ocurrence is of words within subgroups of the text data. These two papers need to be cited:

Rudolph et al., NIPS 2017, "Sturctured Embedding Models for Grouped Data": This paper also presents a method for learning embeddings specific for subgroups of the data, but based on hierarchical modeling. An experimental comparison is needed.

Cotterell et al., EACL 2017 "Explaining and Generalizing Skip-Gram through Exponential Family Principal Component Analysis": This paper also derives a tensor factorization based approach for learning word embeddings for different covariates. Here the covariates are morphological tags such as part-of-speech tags of the words.

Due to these two citations, the novelty of both the problem set-up of learning different embeddings for each covariate and the novelty of the tensor factorization based model are limited.

The writing is ok. I appreciated the set-up of the introduction with the two questions. However, the questions themselves could have been formulated differently:
Q1: the way Q1 is formulated makes it sound like the covariates could be both discrete and continuous while the method presented later in the paper is only for discrete covariates (i.e. group structure of the data).
Q2: The authors mention topic alignment without specifying what the topics are aligned to. It would be clearer if they stated explicitly that the alignment is between covariate-specific embeddings. It is also distracting that they call the embedding dimensions topics.
Also, why highlight the problem of authorship attribution of Shakespear's work in the introduction, if that problem is not addressed later on?

In the model section, the paragraphs "notation" and "objective function and discussion" are clear.  I also liked the idea of having the section "A geometric view of embeddings and tensor decomposition", but that section needs to be improved. For example, the authors describe RandWalk (Arora et al. 2016) but how their work falls into that framework is unclear.
In the third paragraph, starting with "Therefore we consider a natural extension of this model, ..." it is unclear which model the authors are referring to. (RandWalk or their tensor factorization?).
What are the context vectors in Figure 1?  I am guessing the random walk transitions are the ellipsoids? How are they to be interpreted?

In the last paragraph, beginning with "Note that this is essentially saying...", I don't agree with the argument that the "base embeddings" decompose into independent topics. The dimensions of the base embeddings are some kind of latent attributes and each individual dimension could be used by the model to capture a variety of attributes. There is nothing that prevents the model from using multiple dimensions to capture related structure of the data. Also, the qualitative results in Table 3 do not convince me that the embedding dimensions represent topics. For example "horses" has highest value in embedding dimension 99. It's nearest neighbours in the embedding space (i.e. semantically similar words) will also have high values in coordinate 99. Hence, the apparent semantic coherence in what the authors call "topics".

The authors present multiple qualitative and quantitative evaluations. The clustering by weight (4.1.) is nice and convincing that the model learns something useful. 4.2, the only quantitative analysis was missing some details. Please give references for the evaluation metrics used, for proper credit and so people can look up these tasks. Also, comparison needed to fitting GloVe on the entire corpus (without covariates) and existing methods Rudolph et al. 2017 and Cotterell et al. 2017.
Section 5.2 was nice and so was 5.3. However, for the covariate specific analogies (5.3.) the authors could also analyze word similarities without the analogy component and probably see similar qualitative results. Specifically, they could analyze for a set of query words, what the most similar words are in the embeddings obtained from different subsections of the data.

PROS:
+ nice tensor factorization model for learning word embeddings specific to discrete covariates.
+ the tensor factorization set-up ensures that the embedding dimensions are aligned
+ clustering by weights (4.1) is useful and seems coherent
+ covariate-specific analogies are a creative analysis

CONS:
- problem set-up not novel and existing approach not cited (experimental comparison needed)
- interpretation of embedding dimensions as topics not convincing
- connection to Rand-Walk (Aurora 2016) not stated precisely enough
- quantitative results (Table 1) too little detail:
        * why is this metric appropriate?
        * comparison to GloVe on the entire corpus (not covariate specific)
        * no reference for the metrics used (AP, BLESS, etc.?)
- covariate specific analogies presented confusingly and similar but simpler analysis might be possible by looking at variance in neighbours v_b and v_d without involving v_a and v_c (i.e. don't talk about analogies but about similarities)

---

### Official Review · AnonReviewer2 · 2017-11-27
**This paper proposes a covariate aware tensor embedding for text corpora that learns a shared embedding and how different contexts can modify the embedding. The authors show the method recovers interpretable latent embeddings from two text corpora, however, some of the experimental results seem less convincing.**

**Rating:** 5
**Confidence:** 4

**Review:**

This paper presents an embedding algorithm for text corpora that allows known
covariates, e.g. author information, to modify a shared embedding to take context
into account. The method is an extension of the GloVe method and in the case of
a single covariate value the proposed method reduces to GloVe. The covariate-dependent
embeddings are diagonal scalings of the shared embedding. The authors demonstrate
the method on a corpus of books by various authors and on a corpus of subreddits.
Though not technically difficult, the extension of GloVe to covariate-dependent
embeddings is very interesting and well motivated. Some of the experimental results
do a good job of demonstrating the advantages of the models. However, some of the
experiments are not obvious that the model is really doing a good job.

I have some small qualms with the presentation of the method. First, using the term
"size m" for the number of values that the covariate can take is a bit misleading.
Usually the size of a covariate would be the dimensionality. These would be the same
if the covariate is one hot coded, however, this isn't obvious in the paper right now.
Additionally, v_i and c_k live in R^d, however, it's not really explained what
'd' is, is it the number of 'topics', or something else? Additionally, the functional
form chosen for f() in the objective was chosen to match previous work but with no
explanation as to why that's a reasonable form to choose. Finally, the authors
say toward the end of Section 2 that "A careful comparision shows that this
approximation is precisely that which is implied by equation 4, as desired". This is
cryptic, just show us that this is the case.

Regarding the experiments there needs to be more discussion about how the
different model parameters were determined. The authors say "... and after tuning
our algorithm to emged this dataset, ...", but this isn't enough. What type of
tuning did you do to choose in particular the latent dimensionality and the
learning rate? I will detail concerns for the specific experiments below.

Section 4.1:
- How does held-out data fit into the plot?

Section 4.2:
- For the second embedding, what exactly was the algorithm trained on? Just the
  book, or the whole corpus?
- What is the reader supposed to take away from Table 1? Are higher or lower
  values better? Maybe highlight the best scores for each column.


Section 4.3:
- Many of these distributions don't look sparse.
- There is a terminology problem in this section. Coordinates in a vector are
  not sparse, the vector itself is sparse if there are many zeros, but
  coordinates are either zero or not zero. The authors' use of 'sparse' when
  they mean 'zero' is really confusing.
- Due to the weird sparsity terminology Table 1 is very confusing. Based on how
  the authors use 'sparse' I think that Table 1 shows the fraction of zeros in
  the learned embedding vectors. But if so, then these vectors aren't sparse at all
  as most values are non-zero.

Section 5.1:
- I don't agree with the authors that the topics in Table 3 are interpretable.
  As such, I think it's a reach to claim the model is learning interpretable topics.
  This isn't necessarily a problem, it's fine for models to not do everything well,
  but it's a stretch for the authors to claim that these results are a positive
  aspect of the model. The results in Section 5.2 seem to make a lot of sense and
  show the big contribution of the model.

Section 5.3:
- What is the "a : b :: c : d" notation?

---

### Decision · Program_Chairs · 2018-01-29
**ICLR 2018 Conference Acceptance Decision**

**Decision:**

Reject

**Comment:**

The reviewers agree that this paper provides a sensible mechanism for producing word embeddings that exploit correlating features in the data (e.g. texts written by the same author), but point to other work doing the same thing. The lack of direct comparison in the experimental section is troublesome, although it is entirely possible the authors' were not aware of related work. Unfortunately, the lack of an author response to the reviews makes it hard to see the argument in defense of this paper, and I must recommend rejection.